# Southern Ocean control on atmospheric $CO_2$ changes across late-Pliocene Marine Isotope Stage M2

Suning Hou[1], Leonie Toebrock[1], Mart van der Linden[1], Fleur Rothstegge[1], Martin Ziegler[1], Lucas J. Lourens[1], Peter K. Bijl[1]

[1] Department of Earth Sciences, Utrecht University, Utrecht, 3584 CB, the Netherlands

*Correspondence to*: Suning Hou (s.hou@uu.nl)

**Abstract.** During the Pliocene, atmospheric $CO_2$ concentrations ($pCO_2$) were probably sometimes similar to today's and global average temperature was ~3 °C higher than preindustrial. However, the relationships and phasing between variability in climate and $pCO_2$ on orbital time scales are not well understood. Specifically, questions remain about the nature of a lag of $pCO_2$ relative to benthic foraminiferal $\delta^{18}O$ in the late-Pliocene Marine Isotope Stage (MIS) M2 (3300 ka), which was longer than during the Pleistocene. Here, we present a multi-proxy paleoceanographic reconstruction of the late-Pliocene subtropical/subantarctic zone. New dinoflagellate cyst assemblage data is combined with previously published sea surface temperature reconstructions, to reveal past surface conditions, including latitudinal migrations of the subtropical front (STF) over the late-Pliocene at ODP Site 1168, offshore west Tasmania. We observe strong oceanographic variability at the STF over glacial-interglacial timescales, especially the interval (3320–3260 ka) across MIS M2. By providing tight and independent age constraints from benthic foraminiferal $\delta^{18}O$, we find that, much more than benthic $\delta^{18}O$ or local SST, latitudinal migrations of the STF are tightly coupled to global $pCO_2$ variations across the M2. Specifically, a northerly position of the STF during the MIS M2 deglaciation coincides with generally low $pCO_2$. We postulate that the Southern Ocean $CO_2$ outgassing varied strongly with migrations of the STF, and that this in part accounted for the variability in $pCO_2$ across MIS M2.

## 1 Introduction

As the largest exogenic carbon reservoir on Earth, the ocean plays a pivotal role in regulating Earth's climate, through the balance between $CO_2$ uptake and outgassing (Friedlingstein et al., 2022; Sabine et al., 2004). Upwelling in the polar frontal zone flushes respired $CO_2$ from the deep ocean into the atmosphere (Process 1 in Fig. 1a). This process is proposed to be controlled by shifts in sea ice extent and westerlies over glacial and interglacial climates by ocean-only general circulation models (Toggweiler et al., 2006), which move the latitudinal position of oceanic fronts in the Southern Ocean. However, modelling outputs are not always on agreement with each other (Gottschalk et al., 2019) or with geological reconstructions (Rae et al., 2018; Skinner et al., 2010; Venugopal et al., 2023). Hence, significant uncertainties persist in our understanding of ice-ocean-climate interactions. Moreover, the biological carbon pump absorbs dissolved $CO_2$ and removes it from surface waters via export productivity (Martin, 1990; Martínez-García et al., 2014; Thöle et al., 2019), thereby reducing surface

dissolved inorganic carbon (DIC) which enhances $CO_2$ diffusion from the atmosphere (Process 2, 3 in Fig. 1a; Egleston et al., 2010; Gruber et al., 2023). This process mainly takes place at the boundary between the subantarctic and subtropical zone (SAZ), where ocean surface temperature (which has a negative influence on $CO_2$ uptake), ocean stratification (negative), salinity (negative) and DIC (negative) determine $CO_2$ diffusion. The SAZ is nowadays a major carbon sink as a result of both increased anthropogenic emissions and natural ocean circulation (Gruber et al., 2009). The past decades have seen profound

changes in sea surface temperature (SST), salinity (SSS) and the stratification of the SAZ surface waters (Sabine et al., 2004; Gruber et al., 2023). But how these changes will affect the ability of the ocean to act as climate change mitigator in the coming decades, and the amount of excess $CO_2$ that would consequently remain in the atmosphere is currently uncertain (Gruber et al., 2023). This creates a critical uncertainty in the projections of atmospheric $CO_2$ concentration ($pCO_2$) and the resulting effects on climate and sea level, given emission pathway scenarios (Burton et al., 2023; IPCC, 2019).


Reconstructing Southern Ocean conditions in past deglaciation phases might help in understanding interactions between atmospheric climate and ocean conditions. The late-Pliocene is marked by dominant obliquity-controlled benthic foraminiferal oxygen isotope ($\delta^{18}O_{bf}$) increases that have been interpreted as glaciation/cooling phases (e.g., Tiedemann et al., 1994; Shackleton et al., 1995; Lisiecki and Raymo, 2005). The most prominent of which is the Marine Isotope Stage(MIS)M2

(3300 ka; Keigwin, 1987), the deglaciation of which terminates into the mid-Piacenzian Warm Period (mPWP, 3264–3025 ka). Questions remain on the nature of its forcing, but also whether this event is mostly reflective of deep-ocean cooling or ice volume increase. Antarctic ice-proximal lithological and biomarker records suggest surface cooling and ice advance and therefore ice volume increase is involved (Cook et al., 2013; McKay et al., 2012; Patterson et al., 2014), perhaps also on the Northern Hemisphere as suggested by ice-rafted detritus (Flesche Kleiven et al., 2002). In contrast, bottom water temperature

(BWT; Braaten et al., 2023) and ice sheet (Mas e Braga et al., 2023; Yamane et al., 2015) studies suggest limited ice volume change across M2–mPWP transition.

The subsequent mPWP is the most recent time whereby climate conditions were at times equilibrated to modern-like $pCO_2$ of about 400 parts per million (ppm, CENCO2PIP CONSORTIUM, 2023; De la Vega et al., 2020), although there are

discrepancies between proxies and reconstructions, and few records capture sub-orbital temporal resolution (e.g., Seki et al., 2010). Particularly  MIS KM5c (3205 ka) has been a focus point of study because of the similar orbital and continental configuration as today (Haywood et al., 2020). The Pliocene Model Intercomparison Project Phase 2 (PLIOMIP 2; Haywood et al., 2020) compares an ensemble of numerical models run under similar boundary conditions, to global compilations of proxy data from sediment cores (e.g., of sea surface temperature, SST; McClymont et al., 2020). From these efforts, global

average sea surface temperature (~2.3 °C warmer than pre-industrial; McClymont et al., 2020), equilibrium climate sensitivity to $pCO_2$ (2.6–4.8 °C; Haywood et al., 2020) and increased hydrological cycle (wetter equatorial regions, drier subtropical regions; Han et al., 2021) were reconstructed.

The nature and forcing factors behind the M2–mPWP glacial-interglacial transition (3320-3260 ka) is not well understood. High-resolution $pCO_2$ reconstructions for the late-Pliocene reveal low amplitude variability on orbital time scales (De la Vega et al., 2020), i.e., of similar magnitude as that in the late Pleistocene, but the trends in $pCO2$ and $\delta^{18}O_{bf}$ are not as synchronous as in the Pleistocene. Specifically, while PLIOMIP2 demonstrates that overall high $pCO_2$ in the late-Pliocene is likely responsible for the warmer-than-modern climates (Burton et al., 2023), questions remain on the exact phase relationship between $pCO_2$ change and $\delta^{18}O_{bf}$ across the M2–mPWP transition. Available records seem to suggest that $pCO_2$ lags changes in $\delta^{18}O_{bf}$ and (sub)surface cooling about 10–20 kyr (De La Vega et al., 2020; van der Weijst et al., 2022), or in any case are on these time scales not directly related through climate sensitivity to radiative forcing. We further note that collective knowledge on high-resolution $pCO_2$ change across the M2–mPWP interval is restricted to one record from ODP Site 999, North Atlantic. Mg/Ca- and clumped isotope-based deep-sea cooling also demonstrate a lag relative to $\delta^{18}O_{bf}$ (Braaten et al., 2023). These leave the question open how $pCO_2$, ocean and cryosphere influenced each other over the M2-mPWP transition.

Here we investigate how the surface oceanography of one of the current major ocean carbon sinks, the SAZ, changed through the M2-mPWP transition, and infer the implications for the atmospheric $CO_2$ uptake of the region. We present a multiproxy reconstruction of paleoceanographic conditions from Ocean Drilling Program (ODP) Site 1168 (Fig. 1b), offshore west Tasmania, which is located close to the modern position of the subtropical front (STF) and the centre of the modern subantarctic/subtropical zone. We reconstruct surface ocean conditions based on dinoflagellate cyst assemblages, a microplankton group that is strongly tied to specific ocean surface conditions: SST, SSS and nutrients (Thöle et al., 2023). These strict affinities are applied together with previously published biomarker-based sea surface temperature for a detailed reconstruction of changing oceanographic conditions: the latitudinal migration of the subtropical front through time, which potentially deciphers the delayed $pCO_2$ change with respect to $\delta^{18}O_{bf}$.

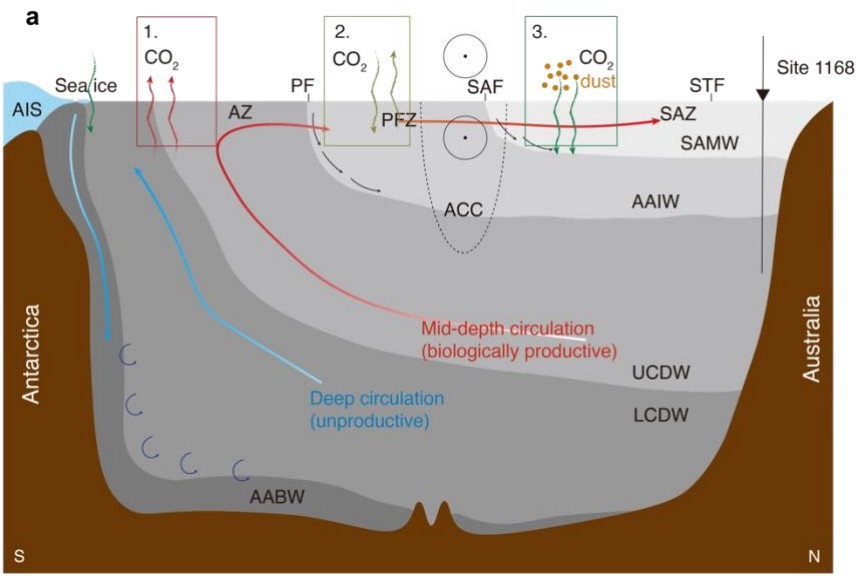

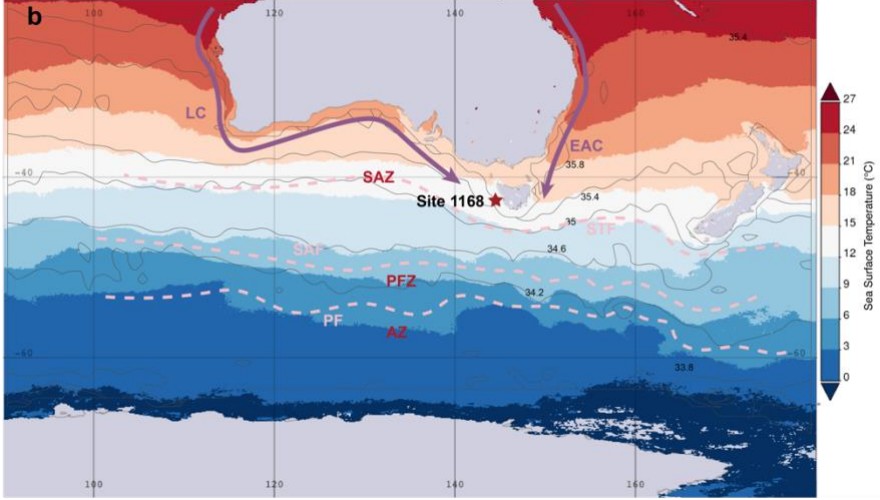

**Figure 1: (a) Schematic view of the ocean circulation in the Southern Ocean between Antarctica and Australia. Arrows in the ocean denote southern overturning circulation (blue), mid-depth overturning circulation (red); grey areas depict water masses; SAMW=subantarctic mode water, AAIW=Antarctic Intermediate Water, U/LCDW=Upper/Lower Component Deep Water, AABW=Antarctic Bottom Water, ACC=Antarctic Circumpolar Current; Curvy arrows denote $CO_2$ uptake or outgassing processes (1. Deep ocean degassing, red; 2. Physical diffusion, spring green; 3. Biological carbon pump, green). (b) Modern site location of ODP Site 1168. Colors indicate sea surface temperatures; Contours indicates sea surface salinity; Grey blocks indicate modern coastline and sea ice extent. Purple arrows denote ocean currents (LC=Leeuwin Current, EAC=East Australia Current). Pink dashed lines denote oceanic fronts (STF=Subtropical Front, SAF=Subantarctic Front, PF=Polar Front) and ocean zones in between (SAZ=Subtropical/Subantarctic Zone, PFZ=Polar Frontal Zone, AZ=Antarctic Zone) are mentioned in red. Data, map and visualization were generated using the Giovanni online data system (https://giovanni.gsfc.nasa.gov/giovanni/) developed and maintained by the National Aeronautics and Space Administration Goddard Earth Sciences Data and Information Services Center (Acker and Leptoukh, 2007). SST and SSS data are derived from Moderate Resolution Imaging Spectroradiometer on the Aqua satellite (MODIS-Aqua) provided to Giovanni by the Ocean Biology Distributed Active Archive Center.**

## 2 Materials and Methods

### 2.1 Study site

ODP Site 1168 (42°36.5809'S, 144°24.7620'E; 2463 meters modern water depth; Fig. 1a) was drilled on the continental slope offshore west Tasmania (Exon et al., 2001). The Pliocene part of the sequence contains greenish-grey foraminifer-bearing nannofossil ooze with significant detrital clay input (Exon et al., 2001). At present, the STF is located closely over this site, which separates warm (>17 °C), saline subtropical waters from comparably cold (<13 °C) and fresh subantarctic water masses (Exon et al., 2001; Heath, 1985). Site 1168 is characterized by a modern SST seasonality ranging from 13–17 °C (winter–summer; Reagan et al., 2023) and a modern BWT of 2.5 °C (Exon et al., 2001).

### 2.2 Palynology

We processed 56 samples for palynology in the late-Pliocene interval. Processing used standard procedures of the GeoLab of Utrecht University (e.g., Brinkhuis et al., 2003). Briefly, this involves first spiking samples with *Lycopodium clavatum* spores prior to palynological processing to allow for quantification of the absolute number of dinocysts per sample (Stockmarr, 1971). Samples were then treated with 30% hydrochloric acid and ~38–40% hydrofluoric acid to concentrate the acid-resistant organic residue. The isolation of the 10-250 μm fraction was established using nylon mesh sieves and an ultrasonic bath to break up agglutinated particles of the residue. Palynomorphs were counted up to a minimum of 200 identified dinocysts if possible. Taxonomy follows that stated on palsys.org (see Bijl and Brinkhuis, 2023; last access 8-1-2024). Functional ecological dinocyst grouping follows those derived from modern assemblages (Fig. 2; Thöle et al., 2023). Notably, *Nematosphaeropsis labyrinthus* is characteristic for the Nlab cluster that prevails south of the STF; *Impagidinium aculeatum*, *Operculodinium centrocarpum* and *Spiniferites* spp. thrive in the Iacu-, high-Ocen-, and Spin- clusters to the north of the STF (Fig. 2). Main taxa are presented in Figure S2. A STF index is then defined as the relative abundance of dinocysts taxa south of the STF (South of STF/(South+North of STF)) in order to quantitatively demonstrate the migration of STF, although the index does not directly indicate the latitudinal position of STF. A higher value of the index indicates that the STF is positioned relatively further north, and vice versa. There are additional dinocysts assemblages specific for Southern Ocean zones further away from the STF (Fig. 2; Thöle et al., 2023). This creates an opportunity to reconstruct in detail past changes in the latitudinal position of the STF through the late-Pliocene, and with that, the oceanographic changes in the subantarctic/subtropical carbon sink (see also Hou et al., 2023b). In addition, given that *Impagidinium pallidum*, which is a typical bipolar cold-water species in the modern ocean (the only *Impagidinium* in the ice-proximal Sant cluster, Fig. 2a), seems to have an ambiguous paleo-affinity (De Schepper et al., 2011) and generally low abundance and widespread occurrence in the modern Southern Ocean (Thöle et al., 2023), it is not separated from the other *Impagidinium* in the grouping. Moreover, because the latitudinal position of the STF is representative of the oceanographic fronts associated with ACC and has implications for the sea ice extent further south, our reconstructions also have implications for the ability of the polar frontal zone to emit $CO_2$ to the atmosphere.

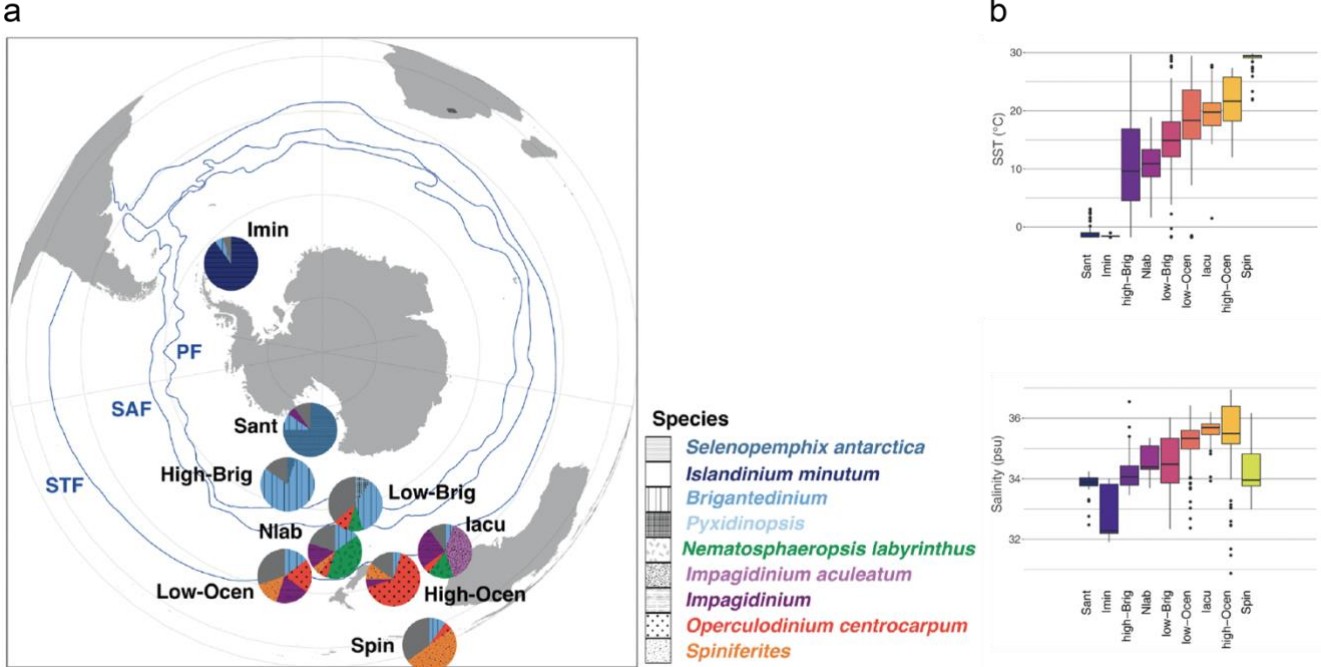

**Figure 2: (a). Schematic representation of the generalized biogeographic distribution of dinocysts in Southern Ocean surface sediments. Pies represent average assemblage composition of the nine clusters described in this paper. Position of these pies represent their typical latitudinal band of occurrence. Also plotted are the frontal systems (blue lines, STF = Subtropical Front, SAF = Subantarctic Front, PF = Polar Front). The Subantarctic Zone (SAZ) is the water mass between the STF and PF. (b) Comparison of sea surface temperature and sea surface salinity in different clusters for the 9-cluster solution of the modern distribution. The median, 25% – 75% quantiles and 95% confidence interval are indicated by the black line, boxes and whiskers, respectively. Modified from Thöle et al. (2023).**

## 2.3 Benthic foraminiferal stable isotopes

Each sediment sample was freeze-dried, washed over a 63 μm sieve, oven-dried at 50 °C and then dry-sieved into different size fractions. We mainly picked tests of *Cibicidoides mundulus* from the 250–355 μm size fraction for our measurements. The picked specimens were cracked between two glass plates after which the test fragments were ultrasonicated in deionized water (3*30 s) to remove adhering sediment, organic lining and nannofossils. The test fragments were dried at room temperature overnight. In order to obtain enough material, other benthic species are also processed. We use *Cibicidoides mundulus* and *Cibicidoides (Planulina) wuellerstorfi* for both stable carbon and oxygen.

Stable isotope measurements were performed using a Thermo Scientific MAT 253 Plus and a Thermo Scientific MAT 253 mass spectrometer at the GeoLab of Utrecht University. Both mass spectrometers were coupled to Thermo Fisher Scientific

Kiel IV carbonate preparation devices. $CO_2$ gas was extracted from carbonate samples with phosphoric acid at a reaction temperature of 70°C. Since both instruments are equipped for clumped isotope analysis, a Porapak trap included in each Kiel IV carbonate preparation system was kept at -40°C to remove organic contaminants from the sample gas. Between each run, the Porapak trap was heated at 120°C for at least 1 h for cleaning. Every measurement run included a similar number of samples and 3 carbonate standards (ETH-1, 2, 3) (Kocken et al., 2019). Two additional reference standards (IAEA-C2 and Merck) were

measured in each run to monitor the long-term reproducibility and stability of the instrument. Both the $\delta^{13}C$ and $\delta^{18}O$ values (reported relative to the Vienna Pee Dee Belemnite (VPDB) scale) of IAEA-C2 showed an external reproducibility (standard deviation) of 0.06 ‰.

## 2.4 Bulk carbonate stable isotopes

Bulk carbonate isotopes were measured as additional stratigraphic tool alongside the benthic $\delta^{13}C$ and $\delta^{18}O$. For 118 samples,

between 50–100 μg of powdered sediment was analysed on a Thermo Finnigan GasBench II system, coupled to a Thermo Delta-V mass spectrometer. Homogenized samples were transferred to sealable vials which were flushed with helium for 5 minutes per vial, to remove atmospheric oxygen and carbon. In each run, 65 samples were then treated with $H_3PO_4$ at a temperature of 72°C together with carbonate standards NAXOS (11 times) and IAEA-603 (4 times) for the purpose of calibration. All isotope values are reported against VPDB. Analytical precision, as determined by the SD of NAXOS was

better than 0.08‰ for $\delta^{18}O$ and 0.04‰ for $\delta^{13}C$.

## 3 Results

### 3.1 Stable isotopes and age model

The post-expedition age model of sediments from ODP Site 1168 comprises of biostratigraphic constraints from nannofossils, foraminifera, diatoms and dinoflagellate cysts, paleomagnetic constraints, and for the Pleistocene identifications of marine

isotope stages from benthic foraminiferal isotopes (Stickley et al., 2004). For the Pliocene-Pleistocene part of the record, the paleomagnetic constraints, which come from Hole B, are structurally offset by around 50 m per 1 million years from biostratigraphic datums and Pleistocene marine isotope stages that come from Hole A, even at splice depth (see Stickley et al., 2004). For a high-resolution age model of the late-Pliocene section at Hole A of Site 1168, we generated new benthic foraminiferal and bulk carbonate stable isotope data across the suspected late-Pliocene interval from Hole A and compared

these to the shipboard colour reflectance data (Exon et al., 2001). Cyclicity in both were then compared to orbital cycles seen in the CENOGRID (Westerhold et al., 2020) (Fig. 3). Since all our new data and the stratigraphic constraints except the paleomagnetic reversals derive from Hole A, which yields the longest extent and best recovery, we decided for the purpose of this study to ignore the offset paleomagnetic constraints from Hole B (as published in Stickley et al. (2004) and updated in Hou et al. (2023a)) for now, and recommend that later studies should first revisit the composite depth, stratigraphic correlation

and quality of the magnetic data before including these into the composite age model of the site.

New $\delta^{18}O_{bf}$ and $\delta^{18}O_{bulk}$ between 27–40 meter below sea floor (mbsf) correlates well with colour reflectance, whereby low/high $\delta^{18}O$ correlates to high/low lightness of the sediment (Fig. 3). Both show a conspicuous trough at 35.0–35.5 mbsf, and based on the available biostratigraphic age model constraints, we interpret that to reflect the MIS M2. Tuning the resulting $\delta^{18}O_{bf}$

and colour reflectance record (Exon et al., 2001) to the CENOGRID (Westerhold et al., 2020) resulted in 4 age tie points (Fig. 3; #3, 4, 5, 6) , of which #3 and 4 are based on the largest slope,  and confidence in the stratigraphic position of MIS M2 isotope excursion. A maximum in $\delta^{18}O_{bulk}$ at 30 mbsf is tuned to MIS G20 (Fig. 3; #2) and a minimum at 37 mbsf is tuned to MIS MG3 (Fig. 3; #8). Additional 2 stratigraphic tie points were chosen by tuning the colour reflectance record to CENOGRID stack further up and down-section (Fig. 3; #1, 7). See Table S1 for the stratigraphic tie points in this paper, and the resulting

age model. Linear regression indicates a sedimentation rate of 1.88 cm/kyr (Fig. 4).

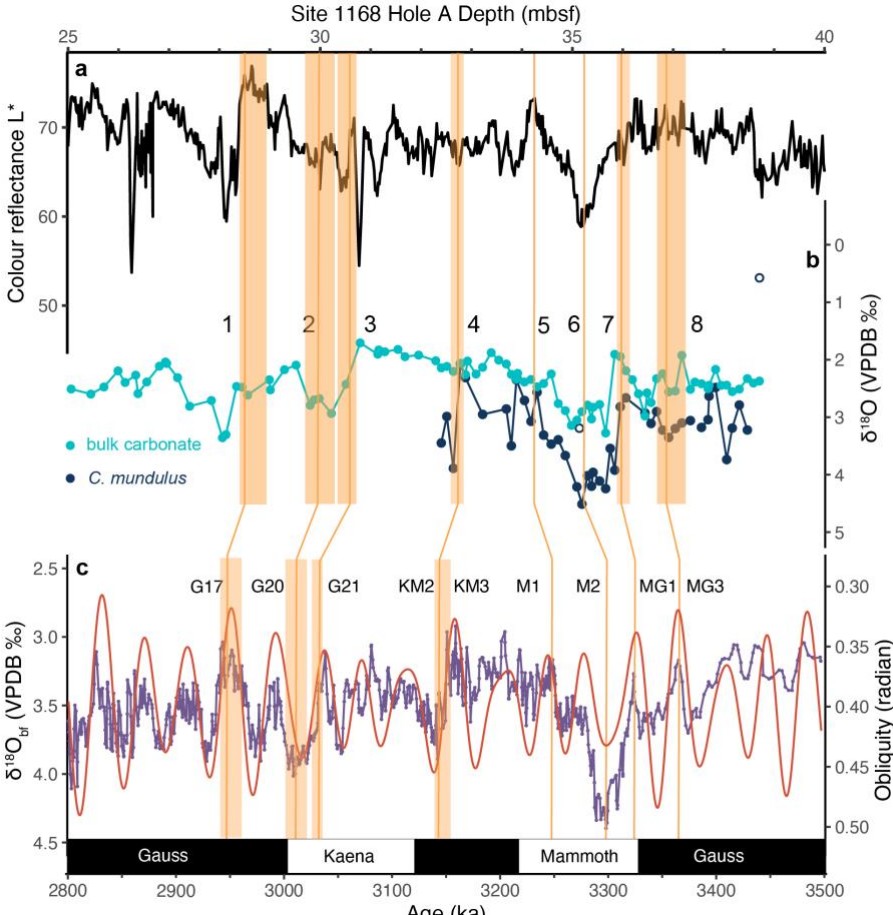

**Figure 3: Age tuning for the Pliocene of ODP Site 1168 Hole A. (a) L\* colour reflectance of Site 1168A (Exon et al., 2001). (b) $\delta^{18}O_{bf}$ and $\delta^{18}O_{bulk}$ of Site 1168A, (c) CENOGRID (Westerhold et al., 2020) and obliquity insolation curve (Laskar et al., 2004) using the software Acycle (Li et al., 2019). Orange lines=tie points; Orange rectangles= errors in depth or age.**

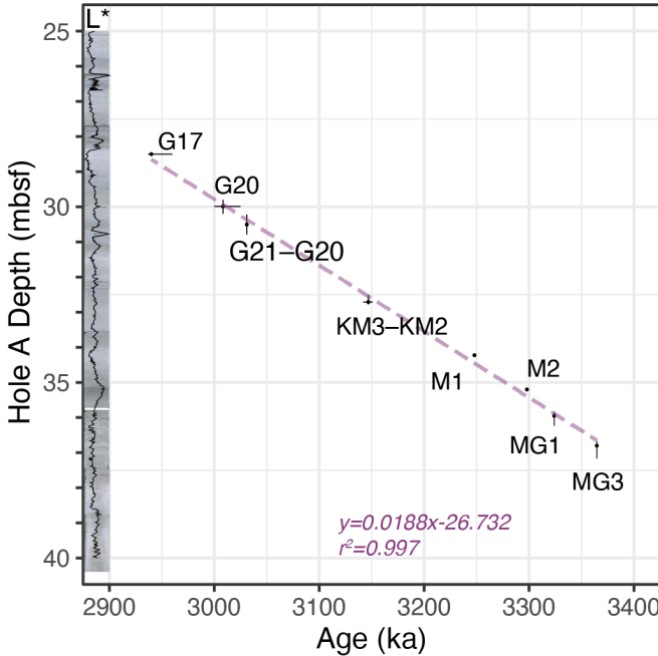

**Figure 4: Age-depth plot of Site 1168 Hole A along with core photos (Core3H6W–Core5H3W) and L\* colour reflectance in the late-Pliocene. Tie points are presented in Fig. 3. Vertical error bars indicate potential errors in depth when tie points are assigned based on $\delta^{18}O_{bf}$ and $\delta^{18}O_{bulk}$, see Table S1; Purple dashed line=linear regression, for an estimation of sedimentation rate, which is used for**
**200    dinocyst burial flux calculation, see supplementary data and Fig. S1.**

### 3.2 Sea surface temperature

SST records of the Pliocene Site 1168 have been previously published (Hou et al., 2023a). SST proxies, $U^{k'}_{37}$ and $TEX_{86}$, were calculated based on alkenones and Glycerol Dialkyl Glycerol Tetraethers (GDGTs) respectively. $U^{k'}_{37}$-based SSTs were determined using core top linear calibration (Müller et al., 1998). $U^{k'}_{37}$-based SSTs vary around 17 °C prior to MIS M2. They
decrease to 12°C at the peak of the MIS M2 glaciation (Fig. 5a). In the mPWP, SST varies around 14 °C, which is approximately 2°C lower than the pre-M2 interval (Fig. 5a). Additionally, SST at KM5c yields 14.5 °C.
$TEX_{86}$-based SSTs are determined by both core top exponential ($TEX_{86}^{H}$; Kim et al., 2010) and Bayesian calibration (BAYSPAR; Tierney and Tingley, 2014). In general $TEX_{86}$-based SSTs resemble those derived from $U^{k'}_{37}$ in trend, however, the amplitude of cooling at MIS M2 is ~3°C higher, which we cannot ascribe to confounding factors in $TEX_{86}$: GDGT-
2/GDGT-3 ratios, a general indicator for additional deep-water contributions to $TEX_{86}$ (Taylor et al., 2013; Ho and Laepple, 2016; van der Weijst et al., 2022), do not change across the MIS M2 (Hou et al., 2023a).

### 3.3 Dinocyst assemblage

Pliocene dinocyst assemblages at Site 1168 are broadly similar to modern assemblages around the subtropical front, thus enable us to use the information of modern affinities of these species (Thöle et al., 2023) to reconstruct paleoceanographic conditions

at this site. Prior to 3400 ka, the STF index is about 0.3 and assemblages are typical for modern regions north of the STF (Fig.

5b), with abundant *O.centrocarpum* (High-Ocen-cluster), *I.aculeatum* (Iacu cluster) and *Spiniferites* spp (Spin-cluster; Thöle et al., 2023). The increase of *N. labyrinthus* (around 3400 ka) makes the assemblages progressively more similar to those of the SAZ, south of the STF and forms the Nlab-cluster when it is dominant in the assemblage (>40%). The attendance of *I. pallidum* is sporadic throughout the record, however, transiently increases to ~10% at 3300ka and dominates the other

*Impagidinium* group (see raw data). The abundance of *N. labyrinthus* peaks at 3275 ka and the STF index reaches 0.8, well after the peak of MIS M2, in its deglaciation stage (Fig. 5b). Thereafter, north-of-STF assemblages recovered and replaced *N. labyrinthus* in the mPWP. Total concentration and total burial flux of dinocyst are generally stable throughout the record but culminate 4-fold at 3240ka (34.05 mbsf; Fig. S1).


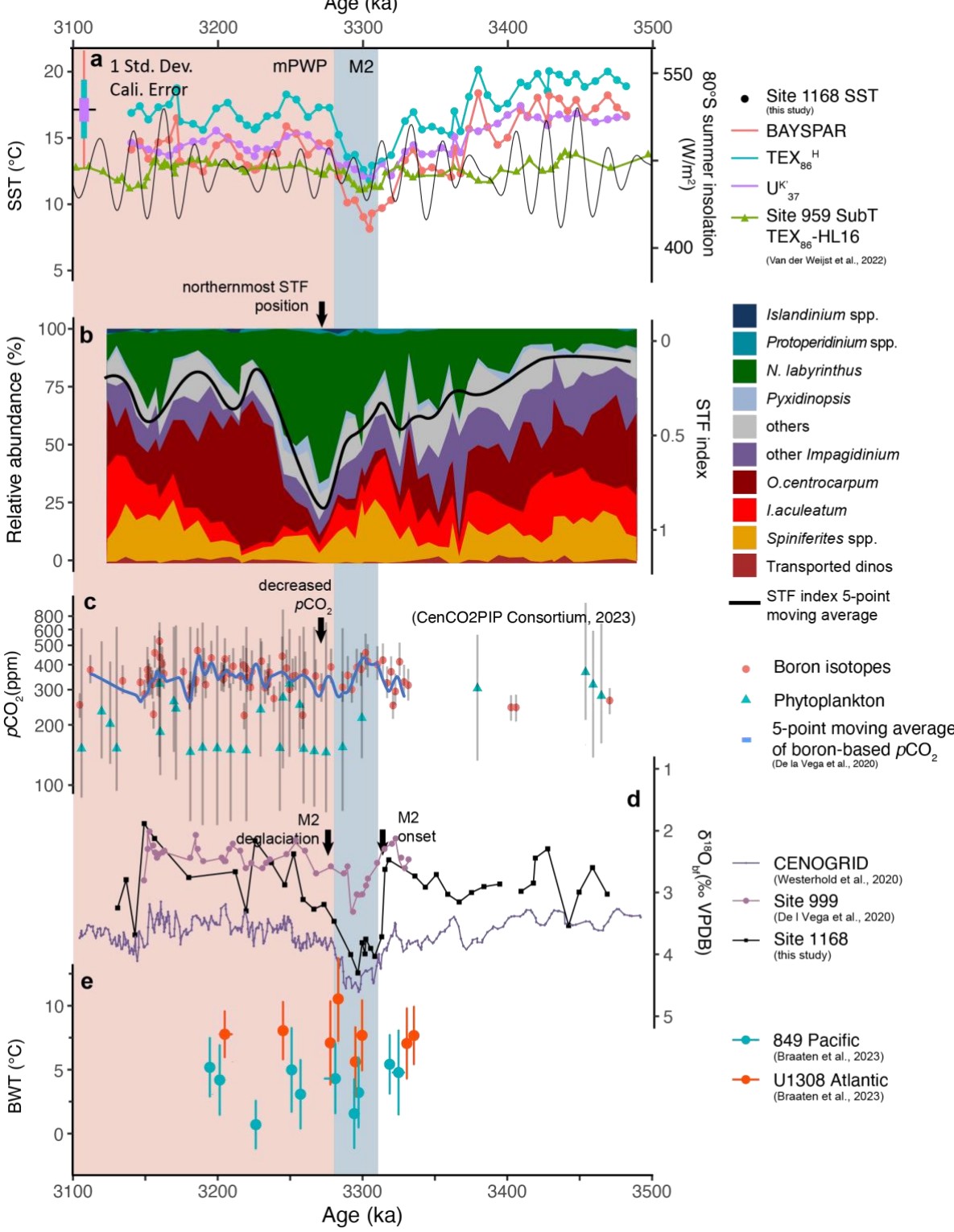

**Figure 5: Late-Pliocene proxy compilation for oceanographic change at ODP Site 1168, and published $pCO_2$, BWT reconstructions. (a)** Sea surface temperature at Site 1168 based on $TEX_{86}$ (exponential $TEX_{86}^H$ and BAYSPAR calibrations; Kim et al., 2010; Tierney and Tingley, 2014) and $U^{k'}_{37}$ (linear calibration; Müller et al., 1998). Subsurface temperature at Site 959 (Van der Weijst et al., 2022) using the HL-16 calibration (Ho and Laepple, 2016). Antarctic summer (80°S January) insolation on the second y-axis (Laskar et al., 2004; de Boer et al., 2014). **(b)** Dinocyst assemblages of Site 1168, green = south of STF species, orange, red and burgundy = north of STF species, petrol and blue = high productivity and/or sea ice affiliated species (Thöle et al., 2023). Black line represents a 5-point moving averge of dinocyst-based STF index (South of STF/(South+North of STF)) roughly indicating the position of the STF at ODP Site 1168 that we derive from these dinocysts assemblages (up or 0 is north, down or 1 is south position). **(c)** $pCO_2$ derived from boron isotopes (red dots) and alkenone $\delta^{13}C$ (cyan triangles) (CENCO2PIP CONSORTIUM, 2023 and references therein) and a 5-point moving average record based on boron isotopes (blue curve); vertical error bar=95% confidence interval. **(d)** Benthic foraminiferal $\delta^{18}O$ of ODP Site 1168, Site 999 (De la Vega et al., 2020) and global stack (Westerhold et al., 2020). **(e)** Bottom water temperature of ODP Site 849 (blue dots) and IODP Site U1308 (orange dots, Braaten et al., 2023), vertical error bar=95% confidence interval, horizontal error bar= averaged age range. Pink rectangle=mid-Piacenzian Warm Period; Blue rectangle=Marine Isotope Stage M2.

## 4 Discussion

### 4.1 STF migrations and SAZ surface conditions in the late Pliocene

Lowest local SSTs (13 °C based on $U^{k'}_{37}$) were recorded at peak MIS M2 glaciation: ~6 °C lower than those before M2 and ~5°C lower than those in the mPWP (Fig. 5a). The amplitude of the SST variation over the mPWP glacial-interglacial cycles is about 1–2 °C, much smaller than the cooling associated with M2. In terms of the cooling amplitude, SSTs in low–mid latitudes during MIS M2 suggest that it represents an unusual strong glacial (Lawrence et al., 2009; De Schepper et al., 2013; Liu et al., 2019, 2022; van der Weijst et al., 2022). However, temperature reconstructions from high latitude surface (Bachem et al., 2017; Risebrobakken et al., 2016) and deep ocean (Braaten et al., 2023) suggest that either MIS M2 indicates no profound cooling, or the cooling has similar amplitude as other glacial phases within the mPWP. The extreme SST response to MIS M2 in the subantarctic zone is therefore extraordinary, and perhaps not the result of radiative forcing but amplified by regional or local oceanographic changes. Furthermore, SSTs of Site 1168 are highly consistent with the subsurface temperature of Site 959 recording South Atlantic Central Water, which derives from the Southern Hemisphere subtropical surface ocean (SACW, van der Weijst et al., 2022). Therefore, their similarity to surface temperatures at Site 1168 is not surprising.

The dinocyst assemblage indicate that the most northern position of the STF is reached during the M2-mPWP transition, i.e. when SST at ODP Site 1168 increased over 5 °C (Fig. 5a, b). During both the peak and deglaciation of MIS M2, SSTs at 1168 are within the modern SST range of Nlab-cluster (Fig. 2), although the 15–17 °C (both proxies) at deglaciation does approach the upper limit of the SST range of Nlab-cluster (Fig. 2b; Thöle et al., 2023). Based on the modern dinocyst distributions (Fig. 2b), and in particular the proliferation of *N. labyrinthus* (Fig. 5b), the surface ocean might have become fresher during MIS M2 deglaciation comparing to pre-M2, according to their modern affinities (Thöle et al., 2023). Since there is no evidence in the palynological slides nor in GDGT-based indices (Hou et al., 2023a) for enhanced terrestrial input from runoff, we infer that the surface ocean freshening of the subantarctic zone at M2 deglaciation originated from excessive iceberg discharge,

which eventually melted in the SAZ or the massive iceberg melting would have impacted larger area than its spatial presence (Merino et al., 2016).

265

Overall, according to the changes we observed in dinocyst assemblages, we estimate that the STF was positioned to the south of Site 1168 from prior to MIS M2 until its onset; the STF moved northward as SST decreased and *N. labyrinthus* increased during M2; During the deglaciation of MIS M2, the STF moved further northward and approached the margin of Tasmania (42°S) at 3275 ka, and surface waters strongly freshened. During the mPWP, the surface salinity at Site 1168 normalized and the STF shifted poleward to a similar position as before M2 (Fig. 5b).

Our interpretation on dinocyst assemblage is mainly based on its modern distribution (Thöle et al., 2023). An evolutionary shift in ecological affinity of dinocyst assemblage/cluster can influence an absolute quantitative estimation of paleo-oceanic conditions. In light of that, modern analogues of dinocyst distribution should be applied with some degree of caution. For example, *Impagidinium pallidum* is restricted to polar regions in modern ocean (Zonneveld et al., 2013), however, it thrived in lower latitudes in the Neogene and associated with higher SSTs (De Schepper et al., 2011; Hennissen et al., 2017). However, the most abundant extant species such as *O. centrocarpum* and *N. labyrinthus* are shown to have comparable SST ranges in the past, by referring to geochemical proxies (De Schepper et al., 2011; Hoem et al., 2021, 2022; Hou et al., 2023b; Sangiorgi et al., 2018), and today. Besides temperature affinities, dinocyst distributions can also indicate salinity in the modern ocean However, quantitative salinity reconstructions remain scarce, and as a result the absolute range of salinities for the Pliocene are unknown. Thus, we can only postulate relative surface salinity change across MIS M2. Given the dinocyst assemblage record found at Site 1168, an alternation from warm (*I. aculeatum* and *O. centrocarpum*) to cool (*N. labyrinthus*) assemblage is distinctive, which was similarly discovered in the Pliocene North Atlantic (De Schepper et al., 2009a, 2011).

## 4.2 Southern Ocean carbon outgasssing as $p$CO$_2$ regulator across M2

By combining our reconstructed STF migrations with the available $p$CO$_2$ reconstructions of the late-Pliocene, we note a coincidence that the northernmost position of the STF is likely synchronous with the lowest $p$CO$_2$, which are both 10–20 kyrs later than MIS M2 (De la Vega et al., 2020). The offset between $p$CO$_2$ from Site 999 and the SST and isotope data shown from Site 1168 is age model independent. Although $\delta^{18}$O records of Site 1168 and Site 999 have demonstrated a reliable stratigraphic match (Fig. 5d), uncertainties remain whether the northernmost position of STF and declined $p$CO$_2$ are really so directly coupled, given the errors in respective age models and the resolution of both records. These give room for a small offset between the STF migration and $p$CO$_2$ decline, but cannot explain the offset between global CO$_2$ and SST at Site 1168. The bulk of the late Pliocene $p$CO$_2$ record is generated from ODP Site 999 (Caribbean Sea), of which the surface air-sea disequilibrium for CO$_2$ is close to 0 (Martínez-Botí et al., 2015). Because of that, this Caribbean Sea site has been used in multiple studies to reconstruct global past $p$CO$_2$ (Chalk et al., 2017; De la Vega et al., 2020, 2023; Foster, 2008).


At the onset of MIS M2, $pCO_2$ was about 400 ppm (De la Vega et al., 2020) and Site 1168 had an abundance of warm species such as *O. centrocarpum*, *I. aculeatum* and *Spiniferites* spp. (Thöle et al., 2023), suggesting a southernly position of the STF. Following this maximum, the STF was moving northwards during MIS M2 $\delta^{18}O$ peak and the coolest SSTs (Fig. 5a). However, The STF reached its northernmost position in the deglaciation phase of M2 event, and this corresponds to the lowest $pCO_2$
(Fig. 5c). During the mPWP, when SST was high, the STF migrated back southward and $pCO_2$ gradually increased to ~400 ppm. The decreased SST and probably salinity should have enhanced the oceanic net uptake of atmospheric $CO_2$ at MIS M2, which had a negative effect on $pCO_2$, however, is contradictory to the high $pCO_2$ reconstructed. Past studies have combined similar dinocyst and SST records across MIS M2 in the North Atlantic, along the path of Atlantic Meridional Overturning Circulation (AMOC; e.g., De Schepper et al., 2009b, 2013, 2014). In those records, no obvious lead-lags can be observed
between dinocyst assemblage, SST and $\delta^{18}O_{bf}$. Such a spatial difference between the North Atlantic (De Schepper et al., 2009b, 2013, 2014) and the Southern Ocean (this study) may be accounted for different forcing processes. Thus, the mechanism we propose involves the ocean as source and sink of atmospheric $CO_2$ (Kirby et al., 2020) and the shifting fronts and Antarctic ice extent (Toggweiler et al., 2006) due to the hysteresis of East Antarctic ice sheet. Our data shows that the two subpolar zones behaved fundamentally differently during the MIS M2 deglaciation phase. Our data is compatible with the hypothesis
proposed by Toggweiler et al. (2006), however, other modelling studies do show opposite results (Gottschalk et al., 2019 and references therein). It should be noted that some feedback mechanisms associated with westerlies/fronts shifts are incompletely represented in models, for instance, Antarctic sea ice cover and ice sheet calving (Gottschalk et al., 2019) and these can seriously impact the outputs. Noteworthily, the consistency of our results with that of Toggweiler et al. (2006) adds to the debate on how oceanography and atmospheric $CO_2$ interact.


The migrations of the oceanic fronts including STF in the Tasmanian sector are the consequences of the shifts in westerlies and Antarctic-proximal sea ice extent – in the Pleistocene and Miocene (Groeneveld et al., 2017; Hou et al., 2023b; Kohfeld and Chase, 2017) but also in the Pliocene. During the M2, the STF gradually shifted northward, indicating an equivalent shift of the westerlies and a northward expansion of the subantarctic zone. The northward migration of the westerlies and fronts
enhanced the stratification of the Southern Ocean and thereby prevented respired $CO_2$ from outgassing into the atmosphere. Consequently, $pCO_2$ dropped, in phase with the northward migration of the STF. At the same time, the freshening of the surface SAZ (Fig. 1) must have lowered carbon uptake in the surface ocean (Bourgeois et al., 2022). However, the decreased $pCO_2$ apparently suggests that the lowered surface carbon uptake did not compensate for the reduction of emission induced by the expanding sea ice cover in the polar frontal zone. The equatorward shift of the STF, which continued into the deglaciation
stage of MIS M2, was associated with expanded sea ice cover in the polar frontal zone, especially in the deglaciation stage, when surface waters freshened. The higher amplitude of obliquity increased Antarctic summer insolation after MIS M2 peak glacial advance (Fig. 5a) and this probably enhanced iceberg calving (De Boer et al., 2014), which stimulated the northward migration and freshening of STF. Furthermore, Antarctic ice sheet simulations suggest that insolation-driven sub-shelf melting can be linked to changes in the carbon cycle (De Boer et al., 2014). Indeed, massive iceberg calving was noticed at the east

Antarctic margin during periods of deglaciation in the Pliocene, associated with maximum iceberg-rafted debris (Cook et al., 2013; Patterson et al., 2014), which is in line with our frontal migration record. Furthermore, geochemical evidences from the North Atlantic exclude the deep Atlantic Ocean as a principle carbon sink, implying a Southern Ocean driving mechanism (Kirby et al., 2020), which is in line with our observations.

When the M2 deglaciation was complete, in the mPWP, iceberg discharge ceased (Patterson et al., 2014) because in the sector of Antarctica nearest to our site fewer glaciers terminated in the ocean (Cook et al., 2013), sea ice cover decreased (Patterson et al., 2014), and westerlies moved southward. As such, shifts in sea ice cover over the polar front controlled air-sea gas exchange: the weaker the sea ice cover, the less stratification, the more $CO_2$ outgassing from the $CO_2$-rich deep water. Similar mechanisms, involving sea ice cover as regulator for Southern Ocean air-sea $CO_2$ exchange, have been proposed for the

Pleistocene and Quaternary (Kohfeld and Chase, 2017; Sigman et al., 2010). Furthermore, the dinocyst-based, poleward positioned STF in the mPWP fell in line with simulated weak stratification and enhanced outgassing in the Southern Ocean (Zhang et al., 2013), which resulted in elevated $pCO_2$. However, new PlioMIP2 models yield contradictory results (Weiffenbach et al., 2023). Simulations on the Southern Ocean thus are highly model dependent (Weiffenbach et al., 2023; Zhang et al., 2021). In any case, present models are not able to resolve frontal migrations or local effects due to their spatial

resolution.

Nevertheless, $pCO_2$ in the Pleistocene (Bereiter et al., 2015; Yan et al., 2019) does not show lags between surface oceanography and benthic $\delta^{18}O$ changes (Chalk et al., 2017; Lisiecki and Raymo, 2005; Martínez-Garcia et al., 2010) as much as the M2-mPWP interval shows here. Shifts in westerlies further drove variations of dust input to the Pleistocene ocean (Abell et al.,

2021) and influenced $CO_2$ uptake through the biological carbon pump (Thöle et al., 2019). Essentially, its impact on carbon storage was in phase with deep ocean $CO_2$ degassing, e.g., inducing lower $pCO_2$ in the Pleistocene glacial maxima (Ai et al., 2020, 2024; Ziegler et al., 2013). However, late-Pliocene aeolian input was limited both regionally in the Southern Ocean (Martínez-Garcia et al., 2010; Naafs et al., 2012) and globally (Teruel et al., 2021), and therefore this process played a less important role during the Pliocene. M2 glaciation occurred mainly as orbital-forced ice buildup and did not seem to have been

triggered by a decline in $pCO_2$ (De la Vega et al., 2020). A new study of $\Delta_{47}$-based BWTs in the north Atlantic and north Pacific has found that deep sea cooling lags the positive $\delta^{18}O$ excursion of M2 by ~20kyrs (Fig. 5d, e; Braaten et al., 2023), but is in phase with the $pCO_2$ variations (De la Vega et al., 2020). Therefore, moderate changes in Pliocene $pCO_2$ across the M2 were independent of global ice volume change but instead linked to oceanographic changes (including deep ocean temperature) through the $pCO_2$-global climate positive feedback (Braaten et al., 2023).

## 5 Conclusions

Our new Pliocene dinocyst assemblage data combined with previously published SSTs from the same site shed new light on the dynamics of Southern Ocean frontal systems, in relation to ice sheet and sea ice. The reconstructions show that the STF migrated substantially across the M2–mPWP climatic transition. Vast sea ice extent and iceberg discharge during the deglaciation stage of MIS M2 pushed the STF to its northernmost position, freshened it, and prevented respired $CO_2$ emissions from the deep ocean to the atmosphere. This suggests that, across MIS M2 event, Southern Ocean frontal migrations controlled ocean-air $CO_2$ exchange and resulted in the $pCO_2$ changes on orbital timescales.

### Data availability

The new palynological and benthic and bulk stable isotope data from Site 1168 are deposited at Zenodo https://doi.org/10.5281/zenodo.8146849. All other data presented have been deposited already, and references to those repository items can be found in the respective publications.

### Author contributions

PKB designed the research. SH and LT processed and analysed samples for palynology, SH and PKB interpreted the palynological results. SH and MvdL washed and picked benthic foraminifera and generated the stable and clumped isotopes data. FR measured the bulk carbonate isotopes. SH, LJL and PKB refined the age model. SH wrote the paper with input from PKB, LJL and MZ. All authors have contributed to the submitted manuscript.

### Competing interests

The contact author declares no competing interests.

### Acknowledgements

We thank Mariska Hoorweg, Natasja Welters, Giovanni Dammers, Desmond Eefting and Arnold van Dijk for laboratory assistance. We thank IODP and scientists of ODP Leg 189, and technicians at KCC in Kochi, Japan for making samples and data available. We are grateful to Lena Thöle, Julia Weiffenbach, Fenghao Liu and Anna Braaten for discussions, and the latter also for providing revised clumped isotope data. This research is funded by ERC Starting Grant 802835 to Peter K. Bijl.

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
