# Peer review of "Southern Ocean control on atmospheric CO2 changes across late-Pliocene Marine Isotope Stage M2"

_Climate of the Past, 2024_

## Referee Comment (RC2)

[referee-annotated manuscript omitted]

---

## Author Response (AR1)

**Reply to RC1**

This paper presents dinoflagellate cyst assemblages from one sediment core (ODP 1168) off the coast of Tasmania across the M2 glacial interval in the Pliocene. By using modern analogs and available SST and CO2 reconstructions it it then concluded how not only SST but also salinity might have changed and how changes relate to CO2. The information is used to suggest that changes in the polar fronts were probably responsible for the observed CO2 changes which are delayed to SST / d18O anomalies across M2.

I am not a data person, so I cannot judge the quality of the data. However, assuming they are ok I try to understand to given story (support for the conclusions) and while I can follow it in principle I have problems in the details. So, therfore, please find my suggestions to improve the text below, which hopefully helps all readers, not only the non-experts, to better follow and understand the content of the paper. Comments are in chronological order.

Reply: We thank the reviewer for the efforts and insights.

1. First sentence of abstract: Todays CO2 are above 420 ppm, while for the Pliocene data are still uncertain, ranging from 150 to 450 ppm, as you also show in Fig 4c. So this needs adjustment, or maybe use a different opening.

Reply: There is consensus that the mPWP has time intervals with elevated pCO2, which is above preindustrial level and around modern values. Certainly, there are errors in reconstructions and discrepancies between proxies, but the late Pliocene has been widely recognised as a high-pCO2 period and modelling work has been conducted by using a ~400ppm pCO2, as suggested by the reviewer below. Proposed changes: "...were **probably sometimes** similar to..."

2. line 11: You show new data from one core. One conclusions is, that the subtropical front migrated several times across the core site, placing the site at some times in the subtropical zone, in other times in the subantarctic zone. So your sentence …of the late-Pliocence subantarctic zone… ist not completely correct.

Reply: As shown in Fig.4b, most of the time the STF is located to the south of Site 1168. Only after M2, the STF moved northward and was located very close to the study site. Obviously the STF can never be north of the site, as Australia is in the way. Nevertheless, a northward push of the STF brings the other fronts closer to the site, and that causes stronger influence of those cold waters towards the site, probably via stronger eddy flow and turbulence.
Proposed changes: "..late-Pliocene **subtropical/**subantarctic zone..."

3. line 11-12: The fact that the subantarctic zone is nowadays a major ocean sink for CO2 comes from the fact that our anthropogenic emissions have reversed the natural atmosphere-ocean CO2 fluxes. However, it makes little sense to compare fluxes of today (fossil fuel driven, so surface ocean CO2 is a slave to atmospheric CO2) with a natural state (atmospheric CO2 is the gas-phase of the ocean, so dominately oceanic-driven atmopsheirc CO2). More interestingly here would be a comparison with the natural role of the Southern Ocean to CO2 (which is a source in the south, see your Fig 1, process 1, and a sink in the north, Fig 1, process 3). Maybe cite for details in the introduction / main text on this subject also Gruber, N., et al. (2009), Oceanic sources, sinks, and transport of atmospheric CO2, Global Biogeochem. Cycles, 23, GB1005, doi:10.1029/2008GB003349.

Reply: We appreciate the insight provided by the reviewer. However, the role of the SAZ as a carbon sink is not only due to anthropogenic emissions, but also a natural result of intermediate water formation from CO2 deprived water which takes up atmospheric CO2.

Proposed changes: The content about the modern day has been removed in the abstract. But we elaborate this in the introduction by adding "**The SAZ is nowadays a major carbon sink as a result of both anthropogenic emissions and natural ocean circulation (Gruber et al., 2009)**":

4. lines 10 and 15: Give the same dates for M2, or better, state it only once at line 10. I do not think ka has to be introduced to the reader of this journal.

Reply: Line 10 indicates the peak glaciation of M2 at 3300ka while line 15 indicates the interval **across M2.**

Proposed changes: We will remove "kiloannum ago".

5. lines 49ff and Fig 4c: On CO2. I would expect a little more critical discussion of CO2. If I look at the CenCO2PIP 2023 data I also find d11B-based CO2 as low as 220 ppm around M2, something which was also highlighted in Köhler (2023, https://doi.org/10.1029/2022PA004439), who only plotted and analysed d11B-based CO2. This together with the even lower alkenone (here called phytoplankton-based) gives me some difficulties in accepting the statement that Pliocene CO2 was ~400 ppm. Maybe yes, maybe not. I can agree, that the de la Vega et al (2020) data are the highest resolved CO2 data across that interval, so the temporal changes are probably real, but the absolute values to some extend also depends on the calibration (and assumptions made for a 2nd variable of the carbonate system, see Köhler 2023 for discussions, if interested). PLIOMIP1 initally used 405ppm, stating explictly that this should also cover changes in non-CO2 greenhouse gases, such as CH4 and N2O. While I think this statement is somehow lost in PLIOMIP2, it is nevertheless contained there, since CH4 and N2O are fixed at preindustrial values. Furthermore, PLIOMIP3 (Haywood et al., 2024, https://doi.org/10.1016/j.gloplacha.2023.104316) now also added a scenario with 490ppm to account for the finding of Hopcroft et al (2020, https://doi.org/10.1073/pnas.2002320117) of higher CH4 in the Pliocene, adding a radiative forcing of 0.9 W/m^2, or a warming of 0.6-1.0 K. Maybe discuss this more widely in your introduction.

Reply: We agree that the absolute values of pCO2 remain uncertain. However, this manuscript focuses on the (timing of) change in the pCO2 across the M2 which is illustrated by both proxies (Fig. 4c), and how it is related to oceanographic changes. But we will tone it down as follows.

Proposed changes: "...400 parts per million (CENCO2PIP CONSORTIUM, 2023; De la Vega et al., 2020), **despite there are discrepancies between different proxies and calibrations.**"

6. line 66: … one record… Please added details, which record this is and where.

Proposed changes: "...one record f**rom ODP Site 999, North Atlantic**."

7. Fig 1: Arrowheads of the CO2 fluxes in Fig 1a are too small. Captions: STZ not explained, while SAZ should be both subtropical and subantarctic zone? I believe the details of the data generation might be better placed in the methods and/or data availability section. What is MODIS-Aqua? I believe there is reference missing for the SST data plotted in Fig 1b.

Proposed changes: We will update the figure and caption as suggested.

8. line 99: Reference missing for modern SST and BWT at site 1168.

Proposed changes: Corresponding references have been added.

9. line 111: The definition of the STF index is crucial for understanding the paper. Please be more specific how this is calculated. How do the dinocysts enter the index. At best state the used equation.

Reply: The equation is already defined as South of STF/(South plus North of STF). Corresponding species have been introduced in the preceding sentences.

Proposed changes: No changes made.

10. line 129: what is the sh_655 data set?

Proposed changes: rephrased as "**modern distribution**".

11. line 147: The sentence „3 carbonate standards (ETH-1,2,3)" is incomplete.

Proposed changes: "Every measurement run included a similar number of samples and 3 carbonate standards (ETH-1, 2, 3) (Kocken et al., 2019). "

12. line 150: „0.06 permil and 0.06 permil, respectively" is better described as „0.06 permil in both variables".

Proposed changes: We will correct as suggest by the reviewer as follows: "**Both the δ13C and δ18O values (reported relative to the Vienna Pee Dee Belemnite (VPDB) scale) of IAEA-C2 showed an external reproducibility (standard deviation) of 0.06 ‰.**"

13. The age model:

 1. On fig 3 you only show d18Obf from CENOGRID, but you mentioned also LR04 in line 178. Is the later really used? If so, maybe include in Fig 3, but consider using the update (Prob-stack, Ahn et al., 2017, https://doi.org/10.1093/climsys/dzx002).

Reply: Given the benthic d18O measured in this study, tying to either record will not lead to a significant difference in the age model or influence our conclusion.

Proposed changes: We will remove the LR04 citation here.

 2. line 179: „4 solid age points": well, which ones are they? there are 7 in Fig 2. The reasoning for chosing the age points is not clear to me, e.g. #7 a minima in CENOGRID d18O is tuned to a maximum in d18O from the site, similarly in #4, or is color reflectance more important here (would make sense for #4, but not for #7), but this is not mentioned in Table 1 as such. Please add details. What about the age model previously used for the published SST data of the site (Hou et al., 2023a)? Is it different, why is this not taken here? Maybe add a depth-age plot.

Reply: the age model used in Hou et al., 2023a does not contain robust constrains from HoleA in the Pliocene, and that study focuses on the long-term trend in the Neogene. In order to better constrain the age on Milanković level and discuss the lead-lag relationship between different records, we measured d18O on both benthic foraminifera and bulk sediments. Tie-point #7 is assumed based on a minimum in d18O **bulk** at 36.8mbsf, as shown in the table. There is indeed a visualization error, and we will fix it.

Proposed changes: "4 age tie points (#3,4,5,6)....MIS G20 (#2)......additional 2 stratigraphic tie points (#1, 7)...."

Proposed changes: We have added a depth-age plot as figure 4.

3. In summary, I have the feeling that the available tie points do not allow for a perfect matching of records. It would be good if you could also state an age uncertainty to the tie points, which would allow to judge how certain the discussed phasing with CO2 is.

Reply: Although uncertainties in the age model of Hole A of Site 1168 remain, we are confident of the assignment of the M2 glaciation, and with that, we can confidently link the dinocysts assemblage change to the pCO2 change. Moreover, our observation that dinocysts assemblage changes are not in phase with both SST change at the same site is completely independent of the age model.

Proposed changes: We have explained the age model more in detail, including the uncertainties involved.

14. Fig 3c: Something is wrong with obliquity (right y-axis), should be around 23°, not around 0.4? Or define precisely, what „obliquity insolation curve" means. Whatever, a unit in the y-axis label would help.

Reply: The angle is represented by radian, which generally comes without a unit in mathematics. It is the default unit whilst generating with Acycle (Li et al., 2019).

Proposed changes: We will add **(radian)** next to axis title.

15. SST, lines 189ff: I have problems to bring the text together with Fig 4a. What I see: there is a SST minima in M2 in BAYSPAR (whatever that is, is not discussed) of 9-10°C, mabye after a 5°C cooling, Pre- and post-M2 SST seem to be on average the same, if you write differently you might need to state more clearly the age window you are refering to.

Reply: We agree with the reviewer's description on the SST, which is, in our opinion, in agreement with our description. But we will clarify both the calibrations and SST results in the text.

Proposed changes:

"**Section 3.2**: SST records of the Pliocene Site 1168 have been previously published (Hou et al., 2023a). SST proxies, Uk'37 and TEX86, were calculated based on alkenones and Glycerol Dialkyl Glycerol Tetraethers (GDGTs) respectively. Uk'37-based SSTs **were determined using core top linear calibration (Müller et al., 1998). Uk'37-based** SSTs vary around 17 °C prior to MIS M2. They decrease to 12°C at the peak of the MIS M2 glaciation (**Fig. 5a**). In the mPWP, SST varies around 14 °C, which is approximately 2°C lower than the pre-M2 interval (Fig. 5a). Additionally, SST at KM5c yields 14.5 °C.

TEX86-based SSTs **are determined by both core top exponential (TEX86H; Kim et al., 2010) and Bayesian calibration (BAYSPAR; Tierney and Tingley, 2014)**. In general **TEX86-based SSTs** resemble those derived from Uk'37 **in trend**, however, the amplitude of cooling at MIS M2 is ~3°C higher, which we cannot ascribe to confounding factors in TEX86: GDGT-2/GDGT-3 ratios, a general indicator for additional deep-water contributions to TEX86 (Taylor et al., 2013; Ho and Laepple, 2016; van der Weijst et al., 2022), do not change across the MIS M2 (**Hou et al., 2023a**)."

16. Fig 4: Please also mark the mPWP in the figure with another vertical bar (as for M2), since you refer to it a lot in the text. You can simplify the legend to Fig 4a: there is only 1 SST from site 959, and that is TEXT86-HL86, so merge green with triangle. All SST records from site 1168 are NOT from „this study" as said in the legend, but if I understood correctly from Hou et al. (2023a). Fig 4c: Maybe highlight

the CO2 data set, that is purely coming from de la Vega et al. (2020), since this was consistently measured on the same site in the same lab. Fig 4a: Why is summer insolation at 80°S plotted, why not at the core site (40°S)?

Reply: We chose to present summer insolation at 80°S based on de Boer et al. (2014), which suggests that Antarctic ice melting is correlated to summer insolation, as in line 281–284.

Proposed changes: Figure updated as suggested.

17. lines 223:ff: SST, again what is described here seems to disagree with Fig 4a, see #15 above. Maybe more details help.

Proposed changes: Please refer to point 15.

18. lines 238ff: I cannot follow why you deduce that salinity changed. Maybe if the STF index is somehow related to SST or SSS I might be able to follow. Or is it simply: SST is already back to preM2 values after the M2, but the assembly distribution is not, therefore it has to be salinity? I can follow that, but how do you get a precise number of 1.5 psu?

Reply: We inferred this from the modern dinocyst distribution model (Thöle et al., 2023), specifically the salinity difference between the modern Nlab cluster and the Iacu cluster, but we agree that the absolute salinity change comes with considerable uncertainty.

Proposed changes: "the surface ocean **might have** become ~1.5 psu fresher during MIS M2 deglaciation **comparing to pre-M2, according to their modern affinities"**

19. line 247: What is the „recovery of M2"? After M2 comes the mPWP, no time lag between both.

Reply: We mean the decrease in d18Obf or the deglaciation stage here.

Proposed changes: Rephrased as "During the **deglaciation** of MIS M2"

20. line 251: I see that front and SST have a phase lag, since both have been measured on the same core. For the lag with CO2 a statement of age uncertainty is in my view helpful.

Reply: Here we are claiming that the STF is mostly synchronous with pCO2, instead of lagging the other proxy. A lag suggests a delayed response, while we question whether they respond to each other at all.

Proposed changes: We have rephrased as follows "By **combining** our reconstructed STF migrations with the available $p$CO$_2$ reconstructions **of the late-Pliocene**, we note a coincidence **that the northernmost position of the STF is likely synchronous with the lowest $p$CO$_2$, which are both 10–20 kyrs later than MIS M2 (De la Vega et al., 2020).**" We will also add δ18O of Site 999 in the last figure.

21. line 263: „influence the ocean uptake effiency of atmospheric carbon". I am not sure this is the right wording. The fronts change the air-sea gas exchange of CO2 and eventually the net CO2 flux. What has efficiency to do with it?

Reply: Here we mean that the increased SST and probably salinity will reduce the ocean's ability of CO2 uptake.

Proposed changes: "**The decreased SST and probably salinity should have enhanced** the ocean uptake efficiency of atmospheric carbon **at MIS M2,** which had an **negative** effect on $p$CO$_2$, **however, is contradictory to the high $p$CO$_2$ reconstructed.**"

22. line 270ff: „... enhanced stratification … preventing respired CO2 from outgassing". I am not sure

you can go that far. Your frontal movement showed, that site 1168 was sometimes north/south of the STF. From that and Fig 1a you can say that the oceanic CO2 uptake (process 3) might be changed, so maybe a smaller marine sink. I do not think you can say a lot more on how the other fronts moved and therefore it is hard to see that the conclusion on stratification and the preventing of outgassing respired CO2 is supported, which would be a smaller marine CO2 source. The net effect on CO2 would be the same but for a different reason.

Reply: We disagree with the reviewer at this point. The migration of the STF is a consequence the ACC and thus westerlies; The equatorward shifted westerlies allow more respired CO2 to accumulate in the deep ocean and the same mechanisms are proposed for the Pleistocene glacial-interglacial cycles. (Toggweiler et al., 2006). Evidence of Antarctic ice dynamics have been provided in lines 280–283. We will elaborate in the discussion.

Proposed changes: We elaborated the discussion where necessary. Please refer to Section 4.2 in the revised manuscript.

23. line 305ff: I am sorry, but I do not see any cooling in BWT in Fig 4e. The data give me more or less a straight line. Furthermore the full citation to Braaten et al (2023) is missing in the reference list, so I cannot check on the details.

Reply: We apologise for the missing reference in the list. The amplitude in BWT can be small. MIS M2 BWT is only 1–2°C cooler than pre-/post- M2, which is the conclusion of the original paper.

Proposed changes: We will update the figure to better illustrate the BWT difference and amend the missing reference.

**Reply to RC2**

Hou and co-authors present dinoflagellate cyst census counts, benthic $\delta^{18}O$ and bulk carbonate stable isotope results obtained from 56 samples covering an interval containing the Pliocene MIS M2 in a single borehole, ODP Site 1168, just west of Tasmania. They complement these results with previously published SST data from the same borehole. They compare their dinoflagellate cyst assemblage compositions to a Southern Hemisphere database detailing the modern distribution pattern of some of the species they encountered in the Pliocene samples. The changes in the composition of the dinoflagellate cyst assemblages are compelling proof of the changing paleoceanographic conditions at ODP Site 1168 during the studied interval. Furthermore, the authors discuss a change in palaeosalinity ("~1.5 psu freshening during the M2 deglaciation"), and a lag $pCO_2$ relative to the benthic foraminiferal $\delta^{18}O$. I find these two findings speculative and I could not find clear evidence for either of those two important claims (see major comments below).

Reply: We thank the reviewer Jan Hennissen for his constructive comments.

**Major Comments**

I believe this is a valuable study as it is located in an area with a notable data shortage in Pliocene paleoceanography in general and MIS M2 in specific. However, I believe there are five points in the current iteration of the manuscript that require addressing:

1) Line 239: 1.5 psu? How did the authors come to that exact number? I was always under the impression only relative palaeo-salinity changes can be derived and not absolute values, like the one stated by the authors. This is because the exact composition of the seawater at the time, the Pliocene,

is unknown (e.g., Rohling, 2007). The authors refer to the presence of *Nematosphaeropsis labyrinthus*, a species which is interpreted as an opportunistic taxon which thrives in rapidly changing oceanic conditions (e.g., Eynaud et al., 2004; Penaud et al., 2008). Presumably, it tolerates a wide range of salinities and may be better adjusted to changing salinities than other taxa. However, is it possible to put an exact number in psu based on the relative abundance changes of a single taxon?

Reply: We are indeed claiming a relative salinity change: Nlab-cluster is characterized by a lower salinity than high-Oper-cluster. However, as RC1 also doubted that based on the modern dinocysts affinities such exact claims could be made, we decided to tone down to a relative salinity change, specifically a freshening at the M2 deglaciation.

Proposed changes: "the surface ocean **might have becom**e ~1.5 psu fresher during MIS M2 deglaciation **comparing to pre-M2, according to their modern affinities**."

2) The authors discuss a lag in $pCO_2$ relative to benthic foraminiferal $\delta^{18}O$, but this is difficult to judge accurately given the unconvincing tie points for the age model presented in Figure 3 and Table 1:The authors match peaks and troughs which is a less objective approach to match sinusoidal curves than using the intervals of highest flux (Martinson et al., 1987; Paillard, 1996) which seems to cause mismatches between their bulk carbonate and their $\delta^{18}O$ from *mundulus* (tie points 3, 5 and 6) where peaks in one curve seem to correlate with troughs in the other and vice versa. Some of the datapoints for $\delta^{18}O$ from *mundulus* during MIS M2 seems to be outliers (unusually low value), are these a true value and are the authors confident it fits the age model? Because of the lack of any error bars this is hard to judge.

Reply: The lag in pCO2 to d18O is found at Site 999 and therefore is not dependent on the age model of Site 1168. As dinocyst data, isotope data and SST data are generated from the same core, the lag of dinocyst assemblage to d18O and SST is not really dependent on the age model either. #3 is a trough in d18Obenthic. #5 indicates troughs in L*, d18Obenthic and d18Obulk. #6 indicates peaks in L*, d18Obenthic and d18Obulk. We are confident on the recognition of the M2 glacial phase. Given the high resolution of L* record, real peaks and troughs can be recognized (#4, 5), thus tying with highest derivative (or slope/or flux as named by the reviewer) would not make huge impact on the phase mismatch between dinocyst and d18Obenthic/SST. But we will try to add slope-based tie-points.

Proposed changes: We will update the age model of Pliocene Site 1168 using slope-based tie-points where possible and present with error bars.

3) The authors directly compare their dinoflagellate cyst assemblages to the modern distribution of some of these taxa. At the very least, some of the caveats in doing this should be acknowledged to inform the reader: This assumes the optimum ecological conditions for the considered taxa has not changed, which we know for at least one taxon, *pallidum,* to be contentious (e.g., De Schepper et al., 2011; Hennissen et al., 2017). Lateral transport and reworking of dinoflagellate cysts has been minimal in the studied samples. Is this so for the studied samples in ODP 1168? When comparing SST records to changes in dinoflagellate cyst assemblages, it is assumed the biotic carrier at the basis of the SST reconstruction shares the same habitat with the dinoflagellates and both the SST biotic carrier and dinoflagellates have the same seasonality.

Reply: We appreciate the concerns raised by the reviewer regarding the ecology of dinocysts. (1). I. pallidum certainly has ambiguous/evolutionary affinities. Given the data generated in the study, the main signal, an alternation between N. labyrinthus and I. aculeatum/O.centrocarpum is clear. Such an

alternation obviously indicates distinctive oceanographic conditions based on both the modern Southern Ocean distribution (Thöle et al., 2023) and Pliocene North Atlantic distribution (De Schepper et al., 2011). (2). Given the location of Site 1168, even if lateral transport happened, the site would receive the dinocysts derived from the same oceanic zone. Reworked dinocysts were not noticed, even though there are sporadic Lingulodinium found. (3) Both SST proxies are calibrated to mean annual SST, so are the dinocyst clusters.

Proposed changes: We will discuss about the potential changes in dinocyst affinities and acknowledge to the suggested literatures in section 4.1 as follows. "**Our interpretation on dinocyst assemblage is mainly based on its modern distribution (Thöle et al., 2023). An evolutionary affinity of dinocyst assemblage/cluster can potential hamper an absolute quantitative estimation of paleo-oceanic conditions. For example, *Impagidinium pallidum* is restricted to polar regions in modern ocean (Zonneveld et al., 2013), however, it thrived in lower latitudes in the Neogene and associated with higher SSTs (De Schepper et al., 2011; Hennissen et al., 2017). Given the dinocyst assemblage record found at Site 1168, an alternation from warm (*I. aculeatum* and *O. centrocarpum*) to cool (*N. labyrinthus*) assemblage is distinctive, which was similarly discovered in the Pliocene North Atlantic (De Schepper et al., 2009, 2011)**."

4) Line 102: "spiking samples with *Lycopodium clavatum*". Why spike the samples if then dinoflagellate cyst concentrations or dinoflagellate cyst burial fluxes are not calculated (I could not find any further results based on the addition of *L. clavatum* , have I missed this?) ? This could be very enlightening as a proxy for palaeoproductivity, especially when the paleoceanography changes as profoundly as the authors suggest. For most IODP/ODP projects the salt free bulk density is measured which in combination with counts of *L. clavatum* and dinoflagellate cyst census counts gives you a dinoflagellate cyst burial flux, a rough indicator for palaeoproductivity corrected for sedimentation rate (e.g., Hennissen et al., 2014; Versteegh et al., 1996)

Reply: Indeed, the concentration/flux was not presented. Total concentration of dinocysts remains relatively stable throughput the record, except a substantial increase at 34.05mbsf (~3240 ka).

Proposed changes: We will update the supplementary data with sample weight, dinocyst concentration and flux. We will incorporate the concentration/flux information into supplementary file and results.

5) This study by Hou et al. shows many parallels to De Schepper et al. (2009) and De Schepper et al. (2013) where a dinoflagellate cyst record is combined with SST measurements and salinity estimates to document paleoceanographic changes over MIS M2. The only difference is the oceanic domain: the current authors focus on the Southern Ocean while De Schepper and co-authors studied the North Atlantic, although they also synthesised their findings in a review paper which includes the Antarctic domain (De Schepper et al., 2014). I find it surprising that the current authors do not compare or contrast their results with those studies given the use of the exact same proxies over the same interval. De Schepper et al. (2013) is only mentioned in passing (line 226) even though those authors clearly articulate potential scenarios for the origins of MIS M2, favouring "a specific forcing, unique within this time period" over astronomical forcing alone.

Reply: We have carefully read these literatures during our study. As the reviewer mentioned, those studies are focusing on the other side of the earth, thus they were not cited in the first submission. Regarding the forcing of MIS M2, it is still mysterious and requires more investigation. Although De

Schepper et al. (2013) has proposed a shallow open Central American Seaway hypothesis, modelling outputs do not support that (Tan et al., 2017; epsl).

Proposed changes: We will cite the suggested literatures, please refer to point 3 above.

**Minor comments**

Line 7: It should be noted that the reason why the Pliocene is an excellent paradigm for end-of-21$^{st}$ century climate is that not only CO2 concentrations and temperatures are comparable to those projected for the end of the century by the IPCC, but certainly as important, especially for a paleoceanographic study, is that the continental configuration was roughly comparable to today's with some important differences in the Panamanian Isthmus that may have regulated ocean currents (e.g., Driscoll and Haug, 1998; Lunt et al., 2008).

Reply: We agree with the reviewer that continental configuration is a crucial additional similarity between modern and the Pliocene.

This study focuses on the reason of pCO2 change across MIS M2 thus we only emphasized the similarity in pCO2.

Proposed changes: "...because of the similar orbital and **continental** configuration as today"

Line 15: I suggest replacing "the M2" with MIS M2 throughout. Also, note the inconsistency here with the date in line 10.

Reply: We have used MIS M2 throughout the manuscript. We have clarified that the latter date indicates the interval across MIS M2, rather than the maximum in d18O.

Proposed changes: Will rephrase as "especially **the interval (3320–3260 ka)** across MIS M2." in the abstract

Line 32: "past decades". Sabine et al. (2004) looked at the period 1800–1994 with some references to the period up to 1999. Since then, nearly three decades passed, invalidating the statement "the past decades". Maybe refer to Gruber et al. (2023)?

Proposed changes: We will cite both.

Line 70: I would add 'current' or 'extant' ahead of ocean carbon sink, because how sure are we it was a major carbon sink during MIS M2?

Proposed changes: We will correct as suggested.

Line 76–77: is it plausible to fix the latitudinal position of an oceanographic front using a single location/borehole?

Proposed changes: We will use "migration" instead of "position".

Lines 169–173: I think this is great to mention why Hole 1168A is preferred over the composite while warning against this practice when unfamiliar with the study material.

Proposed changes: We corrected as follows "...Hole A, **which yields the longest extent and best recovery,** we decided…"

Table 1: Should the source of the first line not be Exon et al., 2001. I understand the tiepoint was chosen by the authors, but they did not generate the data behind it.

Proposed changes: Source of #1 updated

Lines 189–195: why is there such a spread in SST in Figure 4?
Reply: Fig.4a presents the SSTs from both Site1168 and Site 959. SSTs of Site 1168 consist of 2 proxies (UK'37 and TEX86) and 3 calibrations (2 for TEX86).

Line 224: indicate mPWP on Figure 4.
Proposed changes: We will update the figure as suggested.

Line 228: SST from the deep ocean? Rephrase.
Proposed changes: Will rephrase as "temperature reconstructions"

Lines 228–232: I am lost here. The authors state that M2 does not appear to coincide with a major cooling, which is contradicted by their own study (Hou et al., 2023), the results in Figure 4 and what they write in line 229 ("the extreme SST response to M2…").
Reply: Combining previous studies and data in this study, there are spatial differences of the temperature change associated with MIS M2. There is little cooling found in the high latitudes, but the cooling in the mid-latitude (Site 1168) is significant.

Lines 240–242: is this substantiated by for example an increase in IRD in the core?
Reply: We did not accurately investigate the presence of IRD in this core. This site was not really in the likely region of IRD delivery. IRD was not noticed during foraminifera sieving and picking. IRD could have been fully released before the ice bergs entered the SAZ. IRD increase can be found in the Antarctic margin as mentioned in section 4.2 (Cook et al., 2013; Patterson et al., 2014). Furthermore, the impact (freshening/cooling) area of iceberg melting is larger that their spatial presence (Merino et al., 2016; https://doi.org/10.1016/j.ocemod.2016.05.001), Site 1168 could have been affected by meltwater even if no IRD is found.
Proposed changes: We have corrected as "we **infer** that ...which **eventually** melted in the SAZ **or the massive iceberg melting would have impacted larger area than its spatial presence (Merino et al., 2016).**"

Lines 246–247: these appear to be statements of fact not backed up by either observations and calculations or citations. Please amend.
Reply: These are inferred based on our dinocyst data.
Proposed changes: We now refer to Fig. 5b.

Line 258: which species are they? Discuss (with citations) why they are considered to be warm water taxa.
Proposed changes: We have updated as following: "... warm species such as O. centrocarpum, I. aculeatum and Spiniferites spp. (Thöle et al., 2023)…"

Line 313: freshened the sea water, not the STF, right? Rephrase.
Proposed changes: We will correct as suggested by the reviewer.

Just as an aside. When discussing the distribution patterns of dinoflagellates, it would be nice (but not necessary if there are size restrictions) to see a plate depicting the most important dinoflagellate cyst taxa.

Proposed changes: We will add a plate as suggested in the supplementary file.

---

## Referee Report (RR1)

Proposed changes: "the surface ocean **might have becom**e ~1.5 psu fresher during MIS M2 deglaciation **comparing to pre-M2, according to their modern affinities**."

I believe it would be more prudent to drop any reference to any form of salinity units. The problem is that a psu is equal to one gram of salt per 1000 grams of water. As mentioned in my review, we have currently no hard data confirming the exact composition of Pliocene seawater at the psu resolution and most records give relative rather than absolute values.

I believe you make your inferences based on the modern distribution of dinoflagellate cysts in the Southern Hemisphere (Thöle et al., 2023). If the units are to be retained, I would recommend specifically mentioning this caveat at this point and cite the source where modern distribution patterns are linked to extant salinity measurements.

Proposed changes: We will update the age model of Pliocene Site 1168 using slope-based tie-points where possible and present with error bars.

The authors compare their SST, $\delta^{18}O$ and dinoflagellate cyst assemblage composition records from Site 1168 with the pCO2 record from Site 999 and conclude (line 253): "...frontal shifts and $p$CO2 lag SST and benthic $\delta^{18}O$ across M2."

It is true that the dinoflagellate cyst assemblage composition converted to an "STFindex" (line 110) used as a frontal shift indicator lags the benthic isotope composition and that this can be established independent from the age model as they were generated on the core. However, the statement from line 253 above, links it to the pCO2 record from Site 999 and generalises it for MIS M2 and I think it does hinge on a correct age model for the current study.

Equally, the graphic presentation of the data in Figure 4 does, in my opinion, rely on an accurate age model for Site 1168.

I am still not entirely convinced by the age model in its current state (especially from MIS M1 to KM2), but I do agree with the authors that it is very likely that they captured MIS M2, which forms the focus of the study.

Proposed changes: We will discuss about the potential changes in dinocyst affinities and acknowledge to the suggested literatures in section 4.1 as follows. "**Our interpretation on dinocyst assemblage is mainly based on its modern distribution (Thöle et al., 2023). An evolutionary affinity of dinocyst assemblage/cluster can potential hamper an absolute quantitative estimation of paleo-oceanic conditions. For example, *Impagidinium pallidum* is restricted to polar regions in modern ocean (Zonneveld et al., 2013), however, it thrived in lower latitudes in the Neogene and associated with higher SSTs (De Schepper et al., 2011; Hennissen et al., 2017). Given the dinocyst assemblage record found at Site 1168, an alternation from warm (*I. aculeatum* and *O. centrocarpum*) to cool (*N. labyrinthus*) assemblage is distinctive, which was similarly discovered in the Pliocene North Atlantic (De Schepper et al., 2009, 2011)**."

My comment about *I. pallidum* was mainly to serve as an illustration of what could happen if modern analogues are used indiscriminately to interpret palaeontological records. My intention was for this specific example to be included in the current paper, however, I wanted to draw the authors' attention to this assumed ecological uniformitarianism and I believe a broader discussion of the caveats (modern analogues, sharing of ecological niches of the biotic carriers for your SST interpretation and dinoflagellate cysts etc.) is required. This will also address some

of the concerns I expressed in Major Comment #1. I think such a paragraph on caveats could (and probably should) be included in the methodology section or precede the discussion

Reply: Indeed, the concentration/flux was not presented. Total concentration of dinocysts remains relatively stable throughput the record, except a substantial increase at 34.05mbsf (~3240 ka).

Proposed changes: We will update the supplementary data with sample weight, dinocyst concentration and flux. We will incorporate the concentration/flux information into supplementary file and results.

If not included in Figure 4, please do indeed supply it as supplementary data. At 3240 you seem to have your maximum for *Operculodinium centrocarpum* (high-ocean cluster in Thole et al 2023). May be interesting to explore this in the future, but I appreciate this may not be the main focus of the current study.

Reply: We have carefully read these literatures during our study. As the reviewer mentioned, those studies are focusing on the other side of the earth, thus they were not cited in the first submission. Regarding the forcing of MIS M2, it is still mysterious and requires more investigation. Although De Schepper et al. (2013) has proposed a shallow open Central American Seaway hypothesis, modelling outputs do not support that (Tan et al., 2017; epsl).

Proposed changes: We will cite the suggested literatures, please refer to point 3 above.

I agree that the actual records from De Schepper et al. (2009) and (2013) are indeed from the Northern Hemisphere but the mechanisms that these authors propose will have implications for records in the Southern Hemisphere. This is emphasized in De Schepper et al. (2014) where several paragraphs are dedicated to the Antarctic domain.

I agree with the proposed changes for the Minor Comments I raised in the original review.

Thöle, L.M., et al., 2023. An expanded database of Southern Hemisphere surface sediment dinoflagellate cyst assemblages and their oceanographic affinities. J. Micropalaeontol. 42, 35-56 10.5194/jm-42-35-2023.

---

## Author Response (AR2)

This is the 2nd round of review, therefore my main focus was on implemented changes, but I also checked the final draft again.

I can accept most replies and changes made by the authors to the draft. However, I have still a small list of correction, which you find below and one more general point, which I forgot to elaborate on in the 1st round of reviews which is the effect of shifted southern hemisphere westerly winds on CO2, as noted in Toggweiler et al. (2006). The Toggweiler paper is quite controversial discussed in the modelling community for several reasons. One is, that the argument is made in Toggweiler that latitudial shifts in winds might influence CO2, but in the paper only changes in wind strength (no shifts in position) have been investigated with a rather unrealistic model. This and the fact that the model response of CO2 to shifting winds is quite model dependent (even in the sign) are reasons why the Toggweiler et al 2006 should not be cited without further refinements as support for the idea how shifting westerlies might influence CO2. A review of published modelling studies (Gottschalk et al. 2019, doi: https://doi.org/10.1016/j.quascirev.2019.05.013, Fig 14b), found that most studies found a RISE in CO2 for northward shifted winds (and a decrease for southward shift), the opposite of the Toggweiler paper. The effect is small, model-dependent (and different for different climate background conditions) and includes not only a change in the upwelling areas, but subsequently also impacts via changes in nutrients the marine biological carbon pump. For example, Völker & Köhler (2013, doi:10.1002/2013PA002556) analysed the effects of shifting winds in more detail, finding higher CO2 for both southward and northward shifted winds, but for different reasons (how changes in ocean physics and marine biology finally add up to a net effect).

Furthermore, it is only one of many processes which play a crucial role in the carbon cycle. I am aware that here we have a data paper, however the model-based suggestions of Toggweiler should be looked at more critically since different models and groups have not been able to confirm it. Please add some sentences discussing this issue and/or shift from citing mainly the Toggweiler paper to others (e.g. the Gottschalk review paper which is long and difficult to grasp in detail, but looking only at the sections with changes in SHW winds should help, eg section 3.3. there). Or the authors might use this review paper to discuss in more detail other effects related to the paper here, eg Antarctic sea ice (Gottschalk section 3.4).

**Reply: We are very grateful that the reviewer explains the debates in modelling community which helps us better understand the hypothesis and improve the manuscript. We will elaborate our discussion by referring to more modelling paper. Wind/front migration is indeed one of the many process, and other processes have already been discussed in the manuscript.**

**Proposed changes: "Our data is compatible with the hypothesis proposed by Toggweiler et al. (2006), however, other modelling studies do show opposite results (Gottschalk et al., 2019 and references therein). It should be noted that some feedback mechanisms associated with westerlies/fronts shifts are incompletely represented in models, for instance, Antarctic sea ice cover and ice sheet calving (Gottschalk et al., 2019) and these can seriously impact the outputs. Noteworthily, the consistency of our results with that of Toggweiler et al. (2006) adds to the debate on how oceanography and atmospheric CO2 interact."**

Minor comments (line count in the draft with annotated changes):

- line 8: „temperature was ~3°C higher" THAN PREINDUSTRIAL. Without these extra words, it would imply higher than today, which is also about 1.5°C higher than preindustrial, which would add up to 4.5°C higher than preindustrial which I believe is wrong.

**Proposed changes: corrected as suggested**

- line 18: „efficiency of SO carbon outgassing: 2 points: 1) only CO2 can outgass, not carbon; 2) efficiency is in my view a poor choice of words (already mentioned for a different place in 1st round of reviews). I think what is meant here is the „Southern Ocean net CO2 uptake or release"

**Proposed changes: corrected as suggested**

- line 127: I think „Plate S1" should be named „Figure S2"

**Proposed changes: corrected as suggested**

- line 175: „for" missing in „0.04permil d13C"

**Proposed changes: corrected as suggested**

- line 181: 50m PER 1 million years

**Proposed changes: corrected as suggested**

- line 183: „Hole 1158A" -> Hole A of site 1168

**Proposed changes: corrected as suggested**

- line 184: space missing in „HoleA"

**Proposed changes: corrected as suggested**

- lines 200, 203: Still LR04 is mentioned here, but is was argued in the rebuttal that this is not used, please delete, aslo already done on line 186.

**Proposed changes: corrected as suggested**

- lines 200-205: nothing is said on tie point #8

**Proposed changes:** A maximum in $\delta^{18}O_{bulk}$ at 30 mbsf is tuned to MIS G20 (#2) **and a minimum at 37 mbsf is tuned to MIS MG3 (#8).**

- Fig 3: The two right y-axes labels are aligned in different directions, please switch one, at best obliquity to have all y axes labels shwon in the same way

**Proposed changes: corrected as suggested**

- Fig 3 Header: space missing in „HoleA" -> „Hole A"

**Proposed changes: corrected as suggested**

- Fig 3 Caption: „Age tuning of Pliocene Site 1168A" -> „Age tuning for the Pliocene of ODP Site 1168 Hole A"

**Proposed changes: corrected as suggested**

- line 336: Again efficiency in „ocean uptake efficiency of atmospheric carbon". I suggest to rewrite to „oceanic net uptake of atmospheric CO2"

**Proposed changes: corrected as suggested**

- line 339: Typo extant -> extent

**Proposed changes: corrected as suggested**

**Referee #2: Jan Hennissen**

Proposed changes: "the surface ocean **might have becom**e ~1.5 psu fresher during MIS M2 deglaciation **comparing to pre-M2, according to their modern affinities**."

I believe it would be more prudent to drop any reference to any form of salinity units. The problem is that a psu is equal to one gram of salt per 1000 grams of water. As mentioned in my review, we have currently no hard data confirming the exact composition of Pliocene seawater at the psu resolution and most records give relative rather than absolute values.

I believe you make your inferences based on the modern distribution of dinoflagellate cysts in the Southern Hemisphere (Thöle et al., 2023). If the units are to be retained, I would recommend specifically mentioning this caveat at this point and cite the source where modern distribution patterns are linked to extant salinity measurements.

**Reply: We appreciate that the reviewer raised this concern and elaborated once more in the second round.**

**Proposed changes: we removed "1.5psu" and kept the phrasing qualitative.**

Proposed changes: We will update the age model of Pliocene Site 1168 using slope-based tie-points where possible and present with error bars.

The authors compare their SST, δ18O and dinoflagellate cyst assemblage composition records from Site 1168 with the pCO2 record from Site 999 and conclude (line 253): "…frontal shifts and *p*CO2 lag SST and benthic δ18O across M2."

It is true that the dinoflagellate cyst assemblage composition converted to an "STFindex" (line 110) used as a frontal shift indicator lags the benthic isotope composition and that this can be established independent from the age model as they were generated on the core. However, the statement from line 253 above, links it to the pCO2 record from Site 999 and generalises it for MIS M2 and I think it does hinge on a correct age model for the current study.

Equally, the graphic presentation of the data in Figure 4 does, in my opinion, rely on an accurate age model for Site 1168.

I am still not entirely convinced by the age model in its current state (especially from MIS M1 to KM2), but I do agree with the authors that it is very likely that they captured MIS M2, which forms the focus of the study.

**Reply: We appreciate that the reviewer agrees that M2 is captured in the record and the lag between dinocyst and SST is age model independent. The current tie-points yield a robust linear age model and the those associated with MIS M1 and KM2 fall well in line with other tie-points.**

**There could be some wiggle room in how long the lag is, which is dependant on the interpolation method. But the same problem also holds for the pCO2 record of Site 999. Furthermore, Fig. 5d presents the d18O of Site 999 and illustrates the stratigraphic correlation very confidently. There are also more and more evidences that other parts of the earth system, such as deep ocean temperature (Braaten et al., 2022), pCO2 (Kirby et al., 2020) and fronts in the Tasmanian sector (this study) lag d18O.**
**Proposed changes: no changes made.**

Proposed changes: We will discuss about the potential changes in dinocyst affinities and acknowledge to the suggested literatures in section 4.1 as follows. "**Our interpretation on dinocyst assemblage is mainly based on its modern distribution (Thöle et al., 2023). An evolutionary affinity of dinocyst assemblage/cluster can potential hamper an absolute quantitative estimation of paleo-oceanic conditions. For example, *Impagidinium pallidum* is restricted to polar regions in modern ocean (Zonneveld et al., 2013), however, it thrived in lower latitudes in the Neogene and associated with higher SSTs (De Schepper et al., 2011; Hennissen et al., 2017). Given the dinocyst assemblage record found at Site 1168, an alternation from warm (*I. aculeatum* and *O. centrocarpum*) to cool (*N. labyrinthus*) assemblage is distinctive, which was similarly discovered in the Pliocene North Atlantic (De Schepper et al., 2009, 2011).**"

My comment about *I. pallidum* was mainly to serve as an illustration of what could happen if modern analogues are used indiscriminately to interpret palaeontological records. My intention was for this specific example to be included in the current paper, however, I wanted to draw the authors' attention to this assumed ecological uniformitarianism and I believe a broader discussion of the caveats (modern analogues, sharing of ecological niches of the biotic carriers for your SST interpretation and dinoflagellate cysts etc.) is required. This will also address some of the concerns I expressed in Major Comment #1. I think such a paragraph on caveats could (and probably should) be included in the methodology section or precede the discussion

**Proposed changes: We elaborate the discussion as follows: "**...An evolutionary shift in ecological affinity of dinocyst assemblage/cluster can influence an absolute quantitative estimation of paleo-oceanic conditions. **In light of that, modern analogues of dinocyst distribution should be applied with some degree of caution.** For example, *Impagidinium pallidum* is restricted to polar regions in modern ocean (Zonneveld et al., 2013), however, it thrived in lower latitudes in the Neogene and associated with higher SSTs (De Schepper et al., 2011; Hennissen et al., 2017). **However, the most abundant extant species such as *O. centrocarpum* and *N. labyrinthus* are shown to have comparable SST ranges in the past, by referring to geochemical proxies (De Schepper et al., 2011; Hoem et al., 2021, 2022; Hou et al., 2023b; Sangiorgi et al., 2018), and today. Besides temperature affinities, dinocyst distributions can also indicate salinity in the modern ocean However, quantitative salinity reconstructions remain scarce, and as a result the absolute range of salinities for the Pliocene are unknown. Thus, we can only postulate relative surface salinity change across MIS M2.** Given the dinocyst assemblage...**"**

Reply: Indeed, the concentration/flux was not presented. Total concentration of dinocysts remains relatively stable throughput the record, except a substantial increase at 34.05mbsf (~3240 ka).
Proposed changes: We will update the supplementary data with sample weight, dinocyst concentration and flux. We will incorporate the concentration/flux information into supplementary file and results.

If not included in Figure 4, please do indeed supply it as supplementary data. At 3240 you seem to have your maximum for *Operculodinium centrocarpum* (high-ocean cluster in Thole et al 2023). May be interesting to explore this in the future, but I appreciate this may not be the main focus of the current study.

**Reply: We will update the data in zenodo.**

Reply: We have carefully read these literatures during our study. As the reviewer mentioned, those studies are focusing on the other side of the earth, thus they were not cited in the first submission. Regarding the forcing of MIS M2, it is still mysterious and requires more investigation. Although De Schepper et al. (2013) has proposed a shallow open Central American Seaway hypothesis, modelling outputs do not support that (Tan et al., 2017; epsl).

Proposed changes: We will cite the suggested literatures, please refer to point 3 above.

I agree that the actual records from De Schepper et al. (2009) and (2013) are indeed from the Northern Hemisphere but the mechanisms that these authors propose will have implications for records in the Southern Hemisphere. This is emphasized in De Schepper et al. (2014) where several paragraphs are dedicated to the Antarctic domain.

**Reply: we will make a northern-southern hemisphere comparison in section 4.2.**

**Proposed changes: "...**reconstructed. **Previous similar combined dinocyst and SST records across MIS M2 were generated along the path of Atlantic Meridional Overturning Circulation (AMOC; e.g., De Schepper et al., 2009a, 2013, 2014). In those records, no obvious lead-lags can be observed between dinocyst assemblage, SST and $\delta^{18}O_{bf}$. Such a spatial difference may be accounted for different forcing processes.** Thus, the mechanism we propose involves the ocean as source and sink of atmospheric $CO_2$ (Kirby et al., 2020) and the shifting fronts and Antarctic ice extent (Toggweiler et al., 2006) **due to the hysteresis of East Antarctic ice sheet. Our data shows that the two subpolar zones behaved fundamentally differently during the M2 deglaciation phase."**

I agree with the proposed changes for the Minor Comments I raised in the original review.

**Reply: Thank you!**

Thöle, L.M., et al., 2023. An expanded database of Southern Hemisphere surface sediment dinoflagellate cyst assemblages and their oceanographic affinities. J. Micropalaeontol. 42, 35- 56 10.5194/jm-42-35-2023.

---

## Author Response (AR3)

Dear Suning Hue et al.,

**Reply:** actually Suning Hou et al.,

Thanks for submitting your revised manuscript, "Reconciling equatorward migration of Southern Ocean fronts with minor Antarctic ice volume change during Miocene cooling" and "Southern Ocean control on atmospheric CO2 changes across late-Pliocene Marine Isotope Stage M2".

**Reply:** We would like to thank the editor for reviewing and accepting our manuscript. This is the title of the dataset; the manuscript is only the latter one.

I appreciate that you have taken most comments into account.

I have added a few very minor comments to the new version (see attachment).

**Reply:** all corrected accordingly

I also agree with the reviewer questioning the approach to settle age model by tuning to maximum/minimum. Sure, between records in your own core absolute ages is less important. However, a key point here is the relation to the 999 CO2 record. I will ask you to acknowledge the uncertainties associated with the age model(s) and how/if age model uncertainty may affect the results/interpretations.

**Reply:** We have made additional slope-based tie points where possible. Fig.5d has shown the stratigraphic correlation between Site 999 and Site 1168. A small offset between the STF and pco2 is possible but has little effect on our interpretation.

**Proposed changes:**

**Section 4.2 Southern Ocean carbon outgasssing as pCO2 regulator across M2**

By combining our reconstructed STF migrations with the available $pCO_2$ reconstructions of the late-Pliocene, we note a coincidence that the northernmost position of the STF is likely synchronous with the lowest $pCO_2$, which are both 10–20 kyrs later than MIS M2 (De la Vega et al., 2020). **The offset between $pCO_2$ from Site 999 and the SST and isotope data shown from Site 1168 is age model independent. Although $\delta^{18}O$ records of Site 1168 and Site 999 have demonstrated a reliable stratigraphic match (Fig. 5d), uncertainties remain whether the northernmost position of STF and declined $pCO_2$ are really so directly coupled, given the errors in respective age models and the resolution of both records. These give room for a small offset between the STF migration and $pCO_2$ decline, but cannot explain the offset between global $CO_2$ and SST at Site 1168.**

Best regards,

Bjørg Risebrobakken

Editor, Climate of the Past

---

## Author Response (AR4)

Dear Suning Hou and co-authors.

I am happy to accept your manuscript subject to technical correction. In your revised version you state that "The offset between pCO2 from Site 999 and the SST and isotope data shown from Site 1168 is age model independent". Do you mean dependent rather than independent? If so, please correct.

I will look forward to seeing your manuscript published in Climate of the Past.

Best regards,
Bjørg Risebrobakken
Editor, Climate of the Past

Dear editor,
Thank you very much for reviewing and accepting our manuscript. We mean independent in the text; pCO2 lags MIS M2 d18O maximum at Site 999 while lowest SST is coupled to d18O maximum at Site 1168, thus it is age model independent.
Best regards,
Suning Hou and on behalf of co-authors